# Parenteral Exposure of Mice to Ricin Toxin Induces Fatal Hypoglycemia by Cytokine-Mediated Suppression of Hepatic Glucose-6-Phosphatase Expression

**DOI:** 10.3390/toxins14120820

**Published:** 2022-11-23

**Authors:** Seth H. Pincus, Alexi Kyro, Grace A. Maresh, Tami Peters, Jacob Kempa, Tamera K. Marcotte, Zhanguo Gao, Jianping Ye, Valérie Copié, Kejing Song

**Affiliations:** 1Department of Chemistry & Biochemistry, Animal Resource Center (TM), Montana State University, Bozeman, MT 59717, USA; 2Children’s Hospital of New Orleans, New Orleans, LA 70118, USA; 3Departments of Pediatrics and of Microbiology, Immunology and Parasitology, LSU School of Medicine, New Orleans, LA 70112, USA; 4Pennington Biomedical Research Institute, Baton Rouge, LA 70808, USA

**Keywords:** glucose metabolism, hypoglycemia, insulin, ricin, toxin, TNF-α, cytokine induction, β-cell, liver metabolism, glucose-6-phosphatase

## Abstract

Ricin toxin is an agent of biodefense concern and we have been developing countermeasures for ricin threats. In doing so, we sought biomarkers of ricin toxicosis and found that in mice parenteral injection of ricin toxin causes profound hypoglycemia, in the absence of other clinical laboratory abnormalities. We now seek to identify the mechanisms underlying this hypoglycemia. Within the first hours following injection, while still normoglycemic, lymphopenia and pro-inflammatory cytokine secretion were observed, particularly tumor necrosis factor (TNF)-α. The cytokine response evolved over the next day into a complex storm of both pro- and anti-inflammatory cytokines. Evaluation of pancreatic function and histology demonstrated marked islet hypertrophy involving predominantly β-cells, but only mildly elevated levels of insulin secretion, and diminished hepatic insulin signaling. Drops in blood glucose were observed even after destruction of β-cells with streptozotocin. In the liver, we observed a rapid and persistent decrease in the expression of glucose-6-phosphatase (G6Pase) RNA and protein levels, accompanied by a drop in glucose-6-phosphate and increase in glycogen. TNF-α has previously been reported to suppress G6Pase expression. In humans, a genetic deficiency of G6Pase results in glycogen storage disease, type-I (GSD-1), a hallmark of which is potentially fatal hypoglycemia.

## 1. Introduction

Ricin toxin, derived from castor beans, is a prototypical A-B toxin, where the B chain binds promiscuously to cell surface glycans via galactose-specific lectin binding, and the A chain enzymatically inactivates the large ribosomal unit, halting protein synthesis, and causing cell death, primarily by apoptosis [1,2,3,4,5,6,7,8,9]. The syndrome that develops following in vivo exposure to ricin is largely dependent upon the route of exposure. Ricin is considered a potential biothreat because castor beans are readily obtainable, the toxin may be easily purified, it is extremely stable, and lethal at doses as low as 15 µg/kg, depending on the route of exposure. [10,11,12,13,14,15,16,17]. Our lab has studied ricin to develop countermeasures to protect against ricin toxicosis [18,19,20]. In the course of these studies, we sought a humane endpoint to replace lethal challenge by searching for biomarkers of ricin toxicosis. We found that following intraperitoneal (ip) injection of ricin, the sole biochemical abnormality observed in the serum of mice was the rapid development of profound hypoglycemia [21]. Hypoglycemia has also been observed, but not commented upon in porcine [22] and human [23] intoxications. We and others have used hypoglycemia as a surrogate marker for ricin toxicosis [18,24,25]. The current studies were undertaken to define the mechanism(s) underlying the development of hypoglycemia.

We have focused on two aspects of the response to parenteral administration of ricin: inflammation and glucose homeostasis via the pancreatic/hepatic axis. The former because a profound inflammatory response is characteristic following various routes of exposure to ricin and other toxins [16,26,27,28,29,30,31,32]. The latter to investigate the causes of hypoglycemia, as well as the physiologic response to mitigate the dropping blood glucose. Changes observed prior to the development of hypoglycemia (<4 h) probably represent the drivers of hypoglycemia, whereas those observed later are likely a mixture of both causes and responses to hypoglycemia. Results demonstrate very early lymphopenia and TNF-α production. Evidence of insulin production was demonstratable both histologically and biochemically, but insulin signaling was suppressed in the liver, and drops in blood glucose were observed following ricin injection in streptozotocin (STZ)-induced diabetes, indicating excess insulin is not the cause. In the liver, an early decrease in mRNA encoding G6Pase was observed, followed by a drop in immunologically detectable enzyme. TNF-α has previously been reported to suppress G6Pase expression [33,34,35]. In humans a genetic deficiency of G6Pase results in GSD-I, a hallmark of which is potentially fatal fasting hypoglycemia [36,37,38]. In this regard, the syndrome induced by ricin administration resembles GSD-I.

## 2. Results

### 2.1. Glycemic Response to Parenteral Injection of Ricin

We have previously shown that mice become hypoglycemic following ip injection of ricin, with progressively lower blood glucose levels as the time post ricin exposure increases [21]. Figure 1A indicates the terminal blood sugar values in Swiss mice sacrificed at different time points after an ip injection of 30 µg/kg of ricin. These animals were used in the experiments described in the remainder of this manuscript. The results confirm our earlier observations showing a time-related drop in blood sugar, with significant drops starting at 8 h post injection. As blood sugars continue to drop with time, levels become life threatening (<25 mg/dL, between 24 and 48 h) indicating that ricin-induced hypoglycemia would be the likely cause of death had the animals not been euthanized. No age or sex-related differences were observed. To determine if hypoglycemia results from localization of ricin within the peritoneal cavity, we administered intravenous (iv) ricin injections. The decline in blood glucose was even more rapid and lethal (Figure 1B). Because ricin has been shown to induce marked inflammatory responses, we tested the effect of ricin injection in immunodeficient NSG mice lacking the common cytokine receptor γ-chain. The degree of hypoglycemia was similar to that observed in immunocompetent Swiss mice (Figure 1C). Because it is possible that ip injection of ricin led to a failure of the mice to feed and was the subsequent cause of the hypoglycemia, the mice were fasted. Although fasted mice were hypoglycemic at 14 and 24 h, the hypoglycemia was less severe in comparison to mice that received ricin at the same timepoint post injection (*p* < 0.001 at 14 and 24 h). To determine if either component of the ricin holotoxin was solely responsible for hypoglycemia, mice received either only the A-chain (Figure 1E) or the B-chain (Figure 1F), and no change in blood glucose was observed.

### 2.2. Tissue Distribution and Persistence of Ricin

The experiments to measure tissue levels of ricin were performed using antigen capture ELISA which detects intact ricin, capturing with an anti-B chain mAb and detecting with an anti-A chain, thus resulting in the detection of only intact holoenzyme with sub-nanogram sensitivity [39]. Ricin peaked in plasma (Figure 2A) and whole blood (Figure 2B) at 2 h and declined rapidly thereafter. Blood levels of ricin were several orders of magnitude greater than those observed in tissues. Levels of ricin in pancreas and liver, critical locations for glucose homeostasis, plateaued at 2–6 h, and declined thereafter (Figure 2C,D). Splenocytes maintained elevated levels of ricin throughout the experiment (Figure 2E), with the levels being higher than observed in the liver or pancreas, and more persistent than in whole blood. These data do not support the idea that hypoglycemia is the result of the specific binding of ricin to hepatocytes and/or pancreatic cells.

### 2.3. Effects of Ricin Administration on Parameters of Immunity and Inflammation

We enumerated and phenotyped splenic white cells by flow cytometry. Viable cells bearing CD3 (T cells), CD14 (macrophage/monocyte lineage) and CD45R (B cells) were progressively depleted with exposure to ricin (Figure 2F), suggesting that red blood cells (RBCs) in the spleen are the major depot of ricin. Exposure to ricin and other toxins has been shown to induce an acute inflammatory response [16,26,27,28,29,30,31,32]. Therefore, we performed multiplex analysis to measure plasma levels of cytokines, inflammatory markers, and other mediators of intercellular signaling (Figure 3). Changes observed prior to the development of hypoglycemia are apt to represent the drivers of hypoglycemia, whereas those observed later represent both causes and responses to hypoglycemia. We observed several distinct patterns of response, although closer examination reveals the unique kinetic pattern of expression for each cytokine. The assignment of these patterns is simply to aid the reader in examining this large set of data. Noted in pink are changes that occur within the first 1–2 h post ricin injection. There were sharp increases in TNFα and interleukin (IL)-10 at 1 h. To confirm the results with TNFα, a technical replicate was performed and shown in the figure. These two cytokines, TNFα pro-inflammatory and IL-10 with pleiotropic effects immunity and inflammation [40], have also been observed to rise early in sepsis and in acute respiratory failure [41,42,43]. IP-10 drops at 1 h, spikes at 2 h, after which it remains low. Others changing early, drop and either rebound, G-CSF, or remain low, MIP-1α and β. Most mediators changed later in the time course. Some decreased with time (orange panel), others increased (green panels) while no discernable pattern was detected in the remaining (blue panels). Factors that decreased with time began dropping at 4 h or later. These included insulin-like growth factor binding proteins 1–3, soluble VEGF receptors 1–3, the pro-allergenic interleukin-5, and soluble IL-6 receptor. Of particular note is that many of the pro-inflammatory cytokines that are thought to drive cytokine storms associated with other conditions, such as IL-1, IL-6, IL-17, IFN-γ, do not rise until the later time points. The relationship between the levels of cytokines and their soluble receptors was complex. In some cases, IL-6, TNFα and VEGF, there was an inverse relationship of receptor and cytokine levels, whereas IL-1 and sIL-1R both rose.

### 2.4. Systemic Metabolic Responses to Ricin Administration

Metabolites and hormones pertaining to glucose metabolism and alternative energy sources were assayed in the plasma (Figure 3 and Figure 4). Lactate was unchanged following ricin injection (Figure 4A). Ketones increased with time after exposure to ricin (Figure 4B), suggesting cellular catabolism and the use of alternative energy sources as mechanisms to maintain homeostasis. Insulin rose after prolonged exposure (Figure 4C), which was also observed, although to a lesser degree, in the multiplex analysis (Figure 3). This suggests the possibility that hypoglycemia resulted from the overproduction of insulin. Surprisingly, glucagon was significantly elevated prior to the development of hypoglycemia at 2 h, but then showed a high degree of variation among individual mice (Figure 4D). We also observed a 2 h increase in glucagon on the multiplex analysis (Figure 3), although it was not significant, and increased variability was present thereafter. Leptin rose at 18 h and 24 h, after hypoglycemia was established (Figure 3). Insulin-like growth factor (IGF) was diminished after 8 h, whereas some IGF binding proteins diminished with time (IGFBPs 1,2,3), while others increased (IGFBPs 6,7).

A glucose tolerance test was administered to the mice at 4 h, 16 h, or 24 h post ricin administration (Figure 5). To account for the differences in initial blood glucose levels resulting from ricin administration, blood glucose was normalized to a baseline value from the measurement taken immediately prior to glucose injection. The mice that were exposed to ricin for 4 h or 16 h were able to clear the glucose load as well as the mice that were not exposed, with the values returning to or near their baseline values at 120 m. However, at 24 h post ricin administration, glucose clearance was impaired (Figure 5A). The area under the curve (AUC) was calculated to compare the groups. The AUC of the mice exposed to ricin for 24 h was larger, but not significantly so (Figure 5B). This observation paired with the observed increase in insulin levels (Figure 4C) suggested insulin resistance 24 h post ricin. To investigate the mechanism of insulin resistance in ricin-mediated hypoglycemia, we performed immunoblotting to determine whether insulin signaling in the liver was affected following ricin exposure (Figure 6). The relative abundance of the phosphorylated to unphosphorylated forms of proteins involved in the signaling pathway were measured to determine the effects of ricin on the kinase phosphorylation cascade initiated by insulin binding to its receptor. There was a time dependent decrease in the phosphorylation of insulin receptor substrate 1 (IRS-1), RAC-alpha serine/threonine-protein kinase/protein kinase B (AKT-1), and glycogen synthase kinase-3 (GSK3- b), with the decrease becoming significant at 24 h and 48 h post ricin. This indicates that insulin signaling was downregulated, thus providing an explanation for the insulin resistance.

To further clarify insulin’s role in post ricin hypoglycemia, the mice were injected with STZ, a chemotherapeutic agent that is specifically cytotoxic to pancreatic β cells and results in a lack of insulin and chemically induced diabetes [44,45]. Blood glucose increased following STZ injection in all mice. After injection of ricin at 30 µg/kg and 15 µg/kg (Figure 7A), the blood glucose level rapidly dropped to a lethal concentration and mice were sacrificed. After injection of ricin 7.5 µg/kg and 2.5 µg/kg (Figure 7B), blood glucose dropped whether or not the mice received STZ. Following the initial drop, blood glucose rose in mice that received STZ. All of the mice that received ricin 7.5 µg/kg developed signs of ricin toxicity and were euthanized, whether or not they were hypoglycemic or were treated with STZ. Ricin (2.5 µg/kg) was injected a second time into the mice that initially received 2.5 µg/kg of ricin. The second ricin injection proved to be lethal, and the mice were sacrificed (Figure 7B). The plasma insulin levels of the mice in Figure 7B were measured to confirm that the STZ effectively suppressed insulin production, which it did. The insulin concentration was much higher in groups that did not receive STZ in comparison to mice that did (Figure 7C). Thus, the observed drop in blood glucose was independent of insulin production. The insulin resistance that developed in concordance with developing hypoglycemia may represent a defense mechanism, preventing any further lowering of the blood sugar by insulin.

### 2.5. Pancreatic Histology and Function

The pancreas plays a key role in glucose homeostasis. Histological evaluation of pancreases removed at different times post ricin revealed a dramatic expansion in the size of the islets of Langerhans (Figure 8A). To quantify this, pancreatic sections (one section per mouse) were examined, a region of interest was created around each islet, and the surface area was calculated. Measurements were made by investigators blinded to the experimental group. Within 6 h post ricin the mean surface area had doubled and continued to grow over time. The large variation in the size of the islets is a function of sectioning and the irregular shape of islets, since some sections may cut the islet at its widest diameter, whereas others cut tangentially at the poles. Despite this wide variation in size, these differences are highly significant. We next sought to determine which cell type was involved in this expansion. Sections were stained with anti-insulin or glucagon antibodies (Abs) (Figure 8B). The β cells were identified to be the major cell type involved. This was further shown using flow cytometry, reflected in increased forward scatter in insulin-containing cells obtained from suspensions of isolated islets (data not shown). We next addressed whether the increase in the size of the islets represented rapid cell division or expansion in the volume of individual β cells. To accomplish this, we counted the nuclei (stained with Hoechst dye) within the islets and saw no difference (0.0092 ± 0.0011 nuclei per µm^2^ pre ricin vs. 0.0083 ± 0.00037 nuclei per µm^2^ 24 h post ricin).

We also examined the expression of genes that control cell division, using quantitative reverse transcriptase PCR of RNA extracted from isolated islet cells (green panel, Figure 9). While there was some variation in expression of RNA encoding cyclin dependent kinases 1 and 2 and cyclin E at different time points, all significant differences indicated less expression of RNA post ricin. We take these results to indicate that the expansion in islet size was not a matter of cell division, but rather resulted from an increase in the volume of individual cells. In addition, we studied the expression of the genes involved in guiding the differentiation of islet cells (blue panel, Figure 9) and those involved in glucose regulation (pink). We studied differentiation genes to determine whether the change in cell size might reflect a changed differentiation state. We found that expression was either unchanged (INSM1 and Rfx6) or variably decreased over time post ricin (Nkx-2.2, Nkx-6.1, Pax6, and Pdx1), giving little indication that de-differentiation had occurred. Analysis of six key genes involved in glucose regulation yielded somewhat surprising data. At 16 h glucagon expression was significantly decreased, despite the developing hypoglycemia. Similarly, insulin expression decreased over time, despite evidence showing modest elevations of plasma insulin at later time points (Figure 3 and Figure 4). Expression of the glucose transporter *GLUT2* was also markedly depressed in a time dependent manner, whereas *GLUT1* and *GLUT4* were unaffected. Enlarged β cells have been reported in a patient with an activating mutation in glucokinase and neonatal hypoglycemia [46], and we observed that the mRNA for glucokinase was moderately upregulated 2–8 h post ricin. Inactivating mutations in *GLUT2* produce Fanconi–Bickel syndrome, which is characterized by fasting hypoglycemia [47]. We observed significant decreases in *GLUT2* expression beginning at 4 h and decreasing thereafter.

### 2.6. Liver Histology and Function

The liver plays a critical role in glucose storage and the regulation of glycemia. Histologic examination of the liver following the administration of ip ricin showed venous and capillary pooling by 24 h and extensive necrosis at 48 h post ricin (Figure 10A). We next examined the expression of the genes involved in hepatic glucose metabolism (Figure 10B). The expression of all of the genes examined, except GLUT1, was downregulated at later time points, after the initiation of hypoglycemia and tissue damage. Notably, the expression of the enzyme G6Pase demonstrated a dramatic drop at the earliest time points, well before the development of hypoglycemia. G6Pase catalyzes the final step in gluconeogenesis, hydrolyzing glucose-6-phosphate to produce glucose, which may then exit the hepatocyte. A failure to produce this enzyme would limit the ability to counteract hypoglycemia by inhibiting mobilization of glycogen stores within the liver and by interfering with gluconeogenesis. In humans a genetic deficiency of G6Pase results in GSD-1, a hallmark of which is potentially fatal fasting hypoglycemia [36,37,38]. We used ELISA to demonstrate that the level of G6Pase drops with time following ricin administration (Figure 10C), although the time course of the drop is not as precipitous as that of mRNA. Paradoxically, glucose-6-phosphate concentration dropped (Figure 10D), suggesting it is shunted elsewhere, perhaps to the pentose phosphate pathway to counteract oxidative stress. Hepatic glycogen increased steadily with time (Figure 10E), which is consistent with it not being utilized to counteract the dropping blood glucose level. 

## 3. Discussion

Ricin toxin has the potential to be used as a weapon of bioterrorism but may also be a therapeutic agent in immunotoxins. Ricin binds promiscuously to cell-surface glycans found on all cells. Therefore, the clinical syndrome associated with ricin toxicosis is highly dependent on the route of exposure, although one commonality is the induction of a rapid and profound inflammatory response [16,28,29,30,31]. We were surprised to observe that hypoglycemia was the sole change in clinical chemistry analyses following parenteral administration of ricin [21]. To understand the mechanism(s) that result in this fatal hypoglycemia, as well as the response to counter this phenomenon, we have examined the systemic, hepatic, and pancreatic regulators of glucose metabolism in mice injected with ricin toxin. Because the observed hypoglycemia did not develop in mice until 6–8 h post injection (Figure 1), we postulate that changes prior to this are primarily in response to ricin, whereas alterations after that time reflect both ricin toxicity and the homeostatic response to the developing hypoglycemia. Early and dramatic responses occurred within the first 2 h post ricin, including alterations of specific cytokines (Figure 3), as well as suppression of hepatic G6Pase expression (Figure 10).

We initially suspected that increased production of insulin or a failure to produce glucagon might be the basis of hypoglycemia. However, we found that levels of insulin did not increase until a late stage, when hypoglycemia was already established (Figure 3 and Figure 4). Paradoxically we observed an early and impressive increase in the size of pancreatic β cells (Figure 8). We also found evidence of insulin resistance through a glucose tolerance test (Figure 5) and by directly measuring insulin signaling (Figure 6). However, most importantly, we have shown the ability of ricin to induce hypoglycemia despite STZ-induced β-cell destruction (Figure 7), suggesting that insulin plays little role, if any, in the development of ricin-induced hypoglycemia. The β-cell hypertrophy likely indicates that the cells are responding to an unidentified signal, perhaps resulting from the developing insulin resistance. The insulin resistance that developed in concordance with developing hypoglycemia may represent a downstream defense response, preventing any further lowering of the blood sugar by insulin. Surprisingly, there was no evidence that the pancreas responded to the developing hypoglycemia by producing glucagon (Figure 3, Figure 4 and Figure 9), and in fact glucagon mRNA levels were lower when the hypoglycemia was at its nadir.

Gene expression analyses have identified several alterations that could contribute to the development of hypoglycemia. The earliest and most consistent change is a ~10–30× drop in mRNA encoding hepatic G6Pase, accompanied by a decrease in immunoreactive G6Pase protein and an increase in hepatic glycogen stores. Unexpectedly, there is a decrease in hepatic glucose-6-phosphate, suggesting that in the absence of G6Pase activity, the substrate is shunted to other pathways rather than accumulating, possibly to the pentose phosphate pathway to generate NADPH to cope with oxidative stress. Our metabolomic analysis of ricin-treated mice, published in a companion manuscript, support this contention [48]. The decrease in hepatic glucokinase expression that occurs at a late stage also contributes to the decreased level of glucose-6-phosphate. The removal of the phosphate by G6Pase is the final step in the release of free glucose into the circulation from hepatic glycogen stores. A defect in the genes encoding G6Pase leads to GSD-I, which is characterized by hypoglycemia, hepatomegaly, hyperlipidemia, hyperuricemia, and lactic acidosis [36,37,38]. G6Pase−/− mice have similar biochemical findings to human patients with GSD-I, except for the absence of lactic acidosis [49], which is comparable to what we have observed. G6Pase hydrolyzes glucose-6-phosphate so that it may enter the bloodstream as glucose. A deficit results in a decrease in the amount of glucose available to move from stores in the liver to other organs in times of low blood glucose. Because a decrease in G6Pase expression is the earliest and most profound alteration in gene expression that we observed, and the resulting syndrome resembles the effect of GSD-I and G6Pase knockout in mice, we believe it to be a primary driver of hypoglycemia following ricin administration.

Other gene expression alterations having a potential impact on glycemic control were also observed. Hepatic glucokinase was markedly suppressed at later time points, whereas pancreatic glucokinase expression was somewhat increased earlier, before it leveled off. Enlarged islets, resulting from β-cell proliferation, and neonatal hypoglycemia have been reported in a patient with an activating mutation in glucokinase [46], however we did not observe evidence of β-cell proliferation. It has been proposed that hepatic glucokinase plays a central role in glucose homeostasis by functioning as a glucose sensor that is induced by a high insulin to glucagon ratio, such as that seen after feeding [50]. Thus, the decrease in insulin signaling that we observed in the liver (Figure 6) could account for the downregulation of glucokinase expression. GLUT2 expression was significantly diminished with time in both the pancreas and liver. GLUT2 is bidirectional and critical for the movement of glucose across cell membranes [51]. Decreased expression contributes to hypoglycemia due to a lack of glucose distribution from the liver to other tissues. This is observed in patients with GSD-XI, Fanconi–Bickel syndrome, which is characterized by a mutation in the gene encoding for GLUT2 rendering it non-functional [47]. In our study, glycogenolysis and gluconeogenesis were observed to be downregulated in the liver. Higher concentrations of glycogen at the same timepoint that the development of hypoglycemia was observed is indicative of a lack of glycogenolysis. Furthermore, the decrease in the expression of phosphoenolpyruvate carboxykinase in the liver (Figure 10), an enzyme involved in the first bypass of gluconeogenesis and catalyzing the conversion of oxaloacetate to phosphoenolpyruvate, results in a decrease of gluconeogenesis.

In addition to alterations in glucose metabolism, we observed a marked cytokine response that resembles a cytokine storm in many, but not all, regards [52]. We observed a rapid and profound depletion of mononuclear leukocytes (Figure 2F) which may serve to initiate the cytokine storm. Lymphoid depletion has previously been observed following in vivo ricin exposure [17,26]. The pro-inflammatory cytokine TNF-α rises rapidly, but other cytokines associated with the development of a cytokine storm rise late, if at all: IL-1, IFN-γ, IL-17, IL-6. In addition to proinflammatory cytokines, plasma levels of chemokines, soluble cytokine receptors, and even anti-inflammatory cytokine levels rise. The anti-inflammatory and immunosuppressive cytokines, as well as soluble cytokine receptors, may serve as a homeostatic response to dampen the inflammation. Most notably IL-10, which can both promote and suppress inflammation [40], rises within an hour, but then falls. A similar relationship between TNF-α and IL-10 has been observed in other inflammatory syndromes [41,42,43].

A key question is whether the inflammatory and hypoglycemic responses are causally related. The relationship of chronic inflammation to type 2 diabetes and insulin resistance is well established [53,54]. Increased concentrations of TNF-α have been shown to induce insulin resistance by increasing phosphorylation of the insulin signaling pathway in animal and cell models [55], although the administration of anti-TNF-α mAbs had little effect in patients with type 2 diabetes. We observed increased TNF-α concentrations and insulin resistance (Figure 5), decreased phosphorylation along the signaling pathway (Figure 6), and of course hypoglycemia rather than hyperglycemia. These data suggest that the effects of TNF-α on glycemic control may differ between acute and chronic inflammatory processes. Reports that TNF-α administration can induce hypoglycemia and that it decreases G6Pase expression by activation of NF-κB provide a link between ricin-induced inflammatory response and hypoglycemia [33,34,35]. Because the Kupffer cells and liver sinusoidal endothelial cells are especially sensitive to ricin’s effects in vivo [56], local synthesis of TNF-α, either directly by these cells or in response to danger signals released upon cellular damage [57,58], may make the hepatocytes especially susceptible to TNF-α effects, including G6Pase downregulation. A protective mechanism that prevents the development of hypoglycemia by countering oxidative inhibition of liver GTPase has been described in an experimental model of sepsis [59]. This mechanism does not appear to offer such protection following ricin injection, although other aspects of ricin-induced hypoglycemia resemble the hypoglycemia observed in sepsis, which has traditionally been considered to result from depletion of hepatic glycogen stores [60].

A plausible mechanism for the development of hypoglycemia following ricin administration is that the toxin induces a rapid and profound inflammatory response including the rapid induction of TNF-α. This in turn downregulates the expression of G6Pase, preventing hepatic gluconeogenesis leading eventually to a syndrome similar to GSD-I, including the development of systemic hypoglycemia. Whether this path would occur in humans is not known. Inflammatory responses in mice and humans are known to broadly differ [61]. At least one clinical report of ricin toxicosis describes hypoglycemia, but does not indicate whether specific measures were taken to control the low blood glucose in this patient [23]. Other clinical reports do not indicate blood glucose, and direct follow-up with the authors failed to elucidate the question. We did not observe hypoglycemia in macaques exposed to aerosolized ricin [16].

It is well established that apoptotic cell-death plays a major role in ricin induced cytotoxicity. In this manuscript, we have focused on evaluating the effects of ricin on inflammation and glucose homeostasis and have not examined apoptosis. The time course of the inflammatory response and hypoglycemia is considerably more rapid than that of apoptosis. We have observed that when ricin is directly applied to cells, blebbing and other morphological changes are observed within minutes, but that it takes 12–18 h for apoptosis to begin [20]. Here, we report changes in the level of TNF- α and G6Pase occurring within 1 h, and alterations in glucose homeostasis begin 4–6 h following ricin administration. Thus, the kinetics suggest that apoptosis is not the primary driver of the observed hypoglycemia. When the fatal drop in blood sugar was prevented by streptozotocin-induced diabetes, tissue damage and cell death led to fatal outcomes 2–3 days later, demonstrating the difference in the kinetics of hypoglycemia and apoptosis in these mice.

In this report, we have evaluated gene expression in the pancreas and liver using mRNA RT-PCR, rather than by measuring protein levels. Messenger RNA levels are more responsive to changes in stimuli, and studying the mRNA expression of multiple genes is experimentally more direct than protein analyses. In some key cases we measured both mRNA and protein. For insulin and glucagon, plasma levels of protein and pancreatic mRNA were measured (Figure 3, Figure 4 and Figure 9), while G6Pase hepatic levels were measured for both (Figure 10). For G6Pase, the mRNA levels dropped more rapidly and severely than the level of enzyme. One concern about measuring mRNA levels in ricin-treated cells is that ricin affects both transcription and translation [8] and thus the quality or the quantity of the mRNA may suffer, affecting the PCR results. We have observed in cell lines (S.H. Pincus, et al., unpublished data) and in the mice reported here that ricin treatment affects neither RNA quantity nor quality (as measured by RIN) until cell death is observed with vital stains. Moreover, all the RT-PCR results are normalized to the housekeeping gene GAPDH. The normalization would account for differences in mRNA resulting from the overall health of the cells.

The in vivo manifestations of ricin toxicity are highly dependent upon the route of administration because of ricin’s promiscuous cell-binding abilities. Nevertheless, there are important commonalities to the response, notably apoptosis and inflammation. That parenteral administration of ricin would lead to a burst of pro-inflammatory cytokines within hours was predictable. That this would lead to fatal hypoglycemia was not.

To further understand the consequences of these changes, we have performed metabolite profiling analyses of the livers of ricin treated-mice and as a control, food-deprived animals. The results show that each treatment induces a metabolically distinct hypoglycemic state [48]. Understanding the metabolic causes, consequences, and response to hypoglycemia has direct relevance for the management of ricin intoxications, diabetes mellitus, and other metabolic syndromes. In the future we propose to extend these analyses to include insulin-induced hypoglycemia, because of the important adverse consequences of hypoglycemia on the long-term management of diabetes mellitus [62,63,64].

## 4. Materials and Methods

### 4.1. Mice

Outbred Swiss (Cr:NIH(S), Charles River Laboratories, Wilmington, MA, USA) mice were studied, including 164 females and 25 males. Outbred mice were chosen to avoid strain-specific genetic effects. The female to male imbalance resulted from different housing requirements. Outbred mice were between 2 and 5 months of age and weighed between 17–40 g. The mice were treated as follows: fasted (12 mice), injected with saline (34 mice) or ricin 30 µg/kg (124 mice), 15 µg/kg (1 mouse), 7.5 µg/kg (6 mice), 2.5µg/kg (6 mice), ricin A-chain 60 µg/kg (6 mice), or ricin B-chain 60 µg/kg (6 mice). Injections were ip (177 mice) or iv (12 mice). NOD.Cg-*Prkdc^scid^ Il2rg^tm1Wjl^*/SzJ (NSG, Jackson Labs, Bar Harbor, ME, USA) are defective in both adaptive and innate immunity, as well as in cytokine signaling. A total of 14 NSG mice were used: 8 females that were 10 months old, 3 females and 3 males that were 3–4 months old. The mice were 18–30 g and were injected with ricin 30 µg/kg. Fifty-six inbred BALB/c mice (colony breeders originated from Jackson Laboratory, and the colony was maintained at Montana State University’s Animal Resource Center), half male and half female, were studied. Inbred mice limit variability and the strain specific inflammatory response is well documented for BALB/c mice [65,66]. All of the mice were 4–11 weeks old, weighing 20–30 g when studied. BALB/c mice were injected with saline (16 mice) or ricin 30 µg/kg (40 mice). All strains of mice were monitored and euthanized at specific time points.Mice were housed in Tecniplast caging West Chester, PA, USA and individually ventilated with HEPA filters and aspen wood chips (Sanichips, P.J. Murphy’s, Montville, NJ, USA). For enrichment, the mice were given a red mouse house (Bioserve, Flemington, NJ, USA), a nestlet (Ancare, Bellmore, NY, USA, and EnviroDry (Shephard Specialty Papers, Amherst, MA, USA). All cage components were autoclaved prior to use. The mice were fed with irradiated PicoLab Rodent Diet 20 (PMI Nutritional, North Arden Hills, MN, USA). The animals had access to water and feed for the entire experiment, unless otherwise noted. The ambient air temperature was 70–72 °F with a minimum of 30% humidity. The light was on from 5 a.m. to 7 p.m. and off for the remainder of the day for a 14/10 cycle. The female mice were housed 4–5 mice per cage and the males 1–4, depending upon age and aggressiveness.

Previous experiments demonstrated the degree of experimental variability [21]. Based on these results, power analysis was performed to determine the group size. It was determined that 6–8 animals per group provides a >90% probability of obtaining statistically significant differences (*p* < 0.05). Specific group sizes are noted in individual experiments.

Animals were bled > 1 d prior to ricin injection into a heparinized tube or a blood glucose test strip. The mice were bled from the saphenous, facial, or submandibular vein, at the discretion of the animal technician. The mice were injected ip, unless otherwise noted, with 10 µL of volume per gram body weight, via a 27 ga 3/8″ needle. The animals were monitored by AALAS-certified laboratory animal technicians every 20–30 min for the initial 2 h post injection, and every 2 h to 14 h thereafter. Any mice demonstrating signs of distress, pain or suffering were euthanized. The specific criteria for early euthanasia were loss of appetite, markedly diminished activity levels, lethargy, neurologic signs, unexpected behaviors, mottled fur, hunching, difficulty breathing, respiratory sounds, or blood glucose < 25 mg/dL.

At the time of sacrifice, a terminal bleed was performed by intracardiac puncture while under anesthesia. Animals were anesthetized using isoflurane by inhalation, dosing was accomplished by adjusting the position of the nose cone, and anesthesia was confirmed by loss of toe withdrawal reflex. Isoflurane was administered for the duration of the intracardiac bleed and continued until euthanasia. Euthanasia was performed by a cervical dislocation and thoracic incision. The liver, skeletal muscle, pancreas, and plasma were then removed from the mice.

### 4.2. Abs, Reagents, and Test Kits

Ricin holotoxin (5 mg/mL, Ricin Agglutinin II (RCA 60)*, L-1090), ricin A chain (L-1190), and ricin B chain (L-1290) were obtained from Vector Laboratories. The ricin holotoxin (5 mg/mL) was in a saline solution containing 10 mM sodium phosphate, 0.15 M NaCl, pH 7.8, 0.08% sodium azide, 5 mM galactose, then diluted further in PBS prior to use. STZ (Sigma, St. Louis, MO USA) was dissolved immediately prior to use in citrate buffer at pH 4.5 to 0.1 M or 0.02 M. Fluorescent Abs to murine cell surface markers were obtained from eBiosciences (San Diego, CA, USA): PE-Cy7-anti-CD45R/B220, PE-hamster anti-CD3e, and APC-rat anti-CD14 (clone rmC5-3). Similarly labeled isotype matched controls were also obtained from eBiosciences. Abs to detect phosphorylated and unphosphorylated members of the insulin signaling pathway were purchased from Santa Cruz Biotechnology: phospho-IRS-1, IRS-1, phospho-AKT, AKT, phospho-GSK-3β and GSK-3 β. Anti-ricin mAbs used in antigen-capture ELISA were produced from hybridoma cell lines, RAC18 (anti-A chain) and RBC11 (anti-B chain), created in our laboratory [18]. Cells were grown in RPMI1640 (Gibco, Billings, MT, USA) and 5% Ig-depleted FCS (HyClone, Logan, UT, USA) and purified on Protein A agarose (Sigma). RAC18 was conjugated to biotin with an EZ-Link Sulfo-NHS-LC-kiut (Thermo-Fisher). Streptavidin conjugated to horse radish peroxidase was obtained from GE Healthcare (Chicago, IL, USA). Abs for the detection of insulin and glucagon by immunostaining were rabbit Abs (Cell Signaling Technology, Beverly, MA, USA, mAb anti-insulin, polyclonal Ab anti-glucagon) and were detected with alkaline phosphatase-conjugated goat anti-rabbit IgG (Thermo-Fisher, Waltham, MA, USA). Binding was visualized using aminoethyl carbazole (Sigma). The kits used to detect specific biochemistry include: the glucagon-HS ELISA kit (Cosmo Bio USA, Carlsbad, CA, USA), L-lactate assay kit I (Eton Bioscience, San Diego, CA, USA), a colorimetric enzyme assay based on the reduction of tetrazolium salt in a NADH-coupled enzyme reaction to formazan, ketone body assay kit (Sigma), which measures 3-hydroxybutyric acid and acetoacetic acid using an enzymatic assay based on 3-hydroxybutyrate dehydrogenase catalyzed reactions, mouse ultrasensitive insulin ELISA kit (ALPCO, Salem, NH, USA), mouse G6Pc (Glucose-6-Phosphatase, Catalytic) ELISA kit (G-Bioscience, St. Louis, MO, USA), glucose-6-phosphate assay kit (Sigma Aldrich, St. Louis, MO, USA), and glycogen assay kit (Sigma Aldrich). The kits were employed following the manufacturers’ instructions and values measured relative to the standards enclosed with the kits.

### 4.3. Blood Glucose

Blood glucose was measured with a handheld glucometer (Accuchek Advantage, Roche Diagnostics Corp. (Indianapolis, IN, USA), and AgaMatrix. Inc. (Salem, NH, USA)). Blood (25 to 50 μL) was loaded onto the appropriate test strip for each glucometer and read 15 s later. Blood glucose was always measured prior to the experimental procedure and immediately prior to sacrifice. In addition, repeated measurements were made on 68 mice, between 1 and 48 h post ricin injection.

### 4.4. Measurement of Ricin in Tissues

An antigen capture ELISA was used to measure ricin holotoxin in the tissues [18,39]. Tissue samples were obtained at necropsy and immediately frozen. The samples were weighed and mixed with a 5× volume of ice-cold PBS 0.1% Triton X-100 (Sigma). The samples were homogenized for 60 s with a handheld tissue homogenizer, while on ice. Following sedimentation, the supernatants were tested for ricin. The wells of a 96-well round-bottom Immulon 2HB plate (Thermo Scientific) were coated with the mAb RBC-11, then blocked with milk buffer (PBS, non-fat dry milk, Tween 20, 10% Thimersol) for >1 d. The plates were rinsed and incubated with supernatants of tissue homogenates or ricin standards overnight, washed, and incubated with biotinylated mAb RAC 18. Following an overnight incubation, the plates were washed and avidin-conjugated HRP was added. Forty-five min later, the plates were washed, and binding was detected by incubation with tetramethylbenzidine (Sigma) for 15 min, then stopping the process with 1 M sulfuric acid, and reading at 450 nm on an automated plate reader (BioTek, Winooski, VT, USA).

### 4.5. Flow Cytometry

Single cell suspensions of splenocytes were prepared and stained immediately post necropsy. The spleens were minced, passed through a nylon mesh (Falcon), and RBCs were removed with RBC lysis buffer (Sigma). The cells were resuspended in PBS/1%BSA/Azide. The cells (5 × 10^5^) were stained in a total volume of 100 µL with an equal mixture of the following Abs: PE-Cy7-anti-CD45R/B220, PE-anti-CD3e, and APC anti-CD14. The cells were stained for 1 h on ice, washed with PBS, and fixed in PBS/2% paraformaldehyde. Analyses were performed on 10,000 cells gated by FSC and SSC using LSRII (BD Biosciences, San Jose, CA, USA). The data were analyzed with FlowJo software. A compensation algorithm was set using single stained cells and fluorescence-matched isotype controls (eBiosciences).

### 4.6. Multiplex Analyses

The plasma obtained from the mice at the time of necropsy was tested for a panel of cytokines, mediators, and hormones using Milliplex Murine Map kits (MilliporeSigma, Burlington, MA, USA). Following the manufacturer’s instructions, recommended dilutions of plasma were mixed with multiplex beads. Beads and plasma were vortexed and centrifuged for 3 min. The following panels of multiplexed beads were used: bone1A (MBN1A-41K), cytokine/chemokine I (MPXMCYTO70KPMX), soluble cytokine receptor (MSCR-42K-PMX), IGF binding protein (MIGFBP-43K), cardiovascular disease I (MCVD1-77AK), cytokine/chemokine II (MPXMCYP2-PMX12), cytokine/chemokine panel III (MPXMCYP3-PMX6), CRP Single Plex (MCVD77K1CRP), IGF-1 single plex (RMIGF187K), endocrine (MENDO-75K), gut hormone (MGT-78K), and TGFß 1,2,3 (TGFB-64K-03). Each sample was run in duplicate. The samples were then assayed using a Bioplex 200 system (BioRad, Hercules, CA, USA). The concentrations of ligand were determined with the standards included in the bead sets.

### 4.7. Glucose Tolerance Test (GTT)

We used a GTT to determine the ability of ricin-treated mice to clear glucose from the blood. A control group contained 6 mice and was injected with saline 24 h prior to GTT. Experimental groups of 4 mice each received ricin (30 µg/kg) 4 h, 6 h or 24 h prior to GTT. All animals were fasted 6 h prior to the beginning of the GTT, and water was allowed. Blood glucose was measured immediately prior to the start of the GTT. The mice were injected ip with 2 g glucose/kg body weight. Blood sugar was measured, using a glucometer as described previously, 15 m, 30 m, 60 m, and 120 m following glucose injection. Because blood sugar dropped with time post ricin injection, the area under the curve was calculated using a baseline normalized to blood sugar immediately prior to ricin injection.

### 4.8. Immunoblotting

Signaling by insulin was examined in the liver by immunoblotting as described elsewhere [55]. The liver was homogenized in cold lysis buffer (1% Nonidet P-40, 50 mM Hepes, pH 7.6, 250 mM NaCl, 10% glycerol, 1 mM EDTA, 20 mM β-glycerophosphate, 1 mM sodium orthovanadate, 1 mM sodium metabisulfite, 1 mM benzamidine hydrochloride, 10 μg/mL leupeptin, 20 μg/mL aprotinin, 1 mM phenylmethylsulfonyl fluoride). The supernatant was used for immunoblotting after centrifugation at 10,000× *g* for 10 min at 4 °C. Total protein (100 μg) in 50 μL of reducing sample buffer was used per lane. The product was resolved in 7% SDS-PAGE and transferred onto a polyvinylidene difluoride membrane for immunoblotting. Blotting of the membrane was conducted in a milk buffer with the first Ab for 1–24 h and HRP-conjugated secondary Ab for 30 m after. We used matched pairs of Abs recognizing the phosphorylated and total amounts of each protein: IRS-1, AKT, and GSK-3β. The phosphorylated signal and total signal strength were quantified with NIH Image J, and the ratio was calculated. The signals of treated samples are expressed as fold change over the untreated sample.

### 4.9. STZ-Induced Diabetes

STZ was used to induce diabetes as described elsewhere [44]. Prior to each STZ treatment, food was removed at 6 a.m. Four hours later, a fasting blood sugar was obtained, and STZ (40 mg/kg) was injected ip and the food was replaced. The mice received 5 daily doses of STZ. Fasting blood glucose was measured daily. When the blood glucose level was >250 mg/dL for 3 consecutive days, the mice were treated with ricin or saline. In the first experiment, three STZ-treated mice received ricin (30 or 15 µg/kg). The second experiment used 12 mice in 4 groups of 3. Two groups received STZ, two did not. The mice were injected with ricin (7.5 or 2.5 µg/kg). Blood glucose was measured twice daily post ricin. The mice were euthanized according to predetermined clinical criteria as outlined in the IACUC protocol and described above.

### 4.10. Microscopic Analyses

The liver and pancreas were fixed in formalin, embedded in paraffin, sectioned and deparaffinized. For pathologic examination, the tissues were stained with H&E. IHC was performed with rabbit anti-insulin or anti-glucagon primary Abs, alkaline phosphatase-conjugated goat anti-rabbit IgG (Invitrogen, Waltham, MA, USA), and aminoethyl carbazole substrate (Sigma). IHC slides were counterstained with hematoxylin. Images were visualized using the built-in widefield capabilities of a Zeiss LSM 510 microscope and analyzed with Axiovision software (Zeiss, White Plains, NY, USA). Islets were identified by morphologic criteria using a 40× objective, a region of interest (ROI) drawn around the circumference of the islet, and measurement of the ROI’s area was calculated by the software.

### 4.11. Isolation of Islets from Intact Pancreas

Pancreatic islets were isolated as described [67,68]. The mice were anesthetized with ketamine/xylazine. Collagenase (type 4, Worthington, 2 mg/mL) in Medium 199 (Gibco) was injected into the pancreatic duct in situ. The pancreas was immediately removed and placed in a glass screw cap vial on ice. The mouse was euthanized. Collagenase solution was added to each of the pancreases, which were placed in a shaking water bath at 37 °C for 15 min. RPMI 1640 medium with 10% FCS was added to stop the digestion, and the vial was placed on ice. The cells were washed centrifugally ×3 in RPMI 1640 + 10% FCS and resuspended in Ficoll-Paque (GE Healthcare) buffer. The cells were shaken to achieve suspension of pellets. RPMI1640 was layered on top of the Ficoll-Paque and centrifuged at 800× *g* for 20 min. The RPMI layer, interface, and the top of Ficoll-Paque layer were removed, washed, and placed in RPMI1640 + 10% FCS in petri dishes. With gentle swirling under a dissecting microscope, the islets were identified and collected. Isolated islets were saved in RNAlater (Qiagen, Hilden, Germany) or teased into single cell suspensions for immediate study.

### 4.12. Reverse Transcriptase, Quantitative PCR (RT-PCR)

Relative changes in gene expression were quantified using real-time RT-PCR. The primers were designed to have similar annealing temperatures and are shown in Table 1. Total RNA was isolated from the liver or isolated pancreatic islets with an RNeasy mini kit (Qiagen) following DNase treatment. Quantitative real-time RT-PCRs were preformed using an iScript one-step RT-PCR kit with SYBR green (Bio-Rad, Hercules, CA, USA) on the iCycler MyiQ single-color real-time PCR detection system (Bio-Rad). A total of 10 ng of RNA was used per reaction and the transcript levels were normalized to GAPDH levels. The relative mRNA expression levels were calculated according to the comparative C_T_ (∆∆C_T_) method as described by the manufacturer (User Bulletin #2, Applied Biosystems, Waltham, MA, USA). All of the reactions were performed in triplicate.

### 4.13. Statistics

All of the statistics and graphs were generated using GraphPad Prism version 8.4.1 for macOS (GraphPad Software). The Shapiro–Wilk test and Brown–Forsythe test were performed on mice included in Figure 1 to validate the use of parametric tests throughout the manuscript. We have previously shown that blood glucose was the only parameter that significantly changed following ricin administration [21]. We reasoned that if the blood glucose data following ricin administration was parametric, the remaining data would be as well. Student’s *t* tests were performed when two groups were compared. When more than two groups were examined, one-way ANOVA with Dunnett’s multiple comparisons test was administered. The significance level was set at 0.05 for all statistical experiments and all of the groups were compared to the control mice. The mean ± SEM are shown in all graphical representations. If no error bars are visible, they are smaller than the symbol indicating the mean. Representative images are shown for tracing the islets of Langerhans, immunoblotting, and tissue staining experiments.

## Figures and Tables

**Figure 1 toxins-14-00820-f001:**
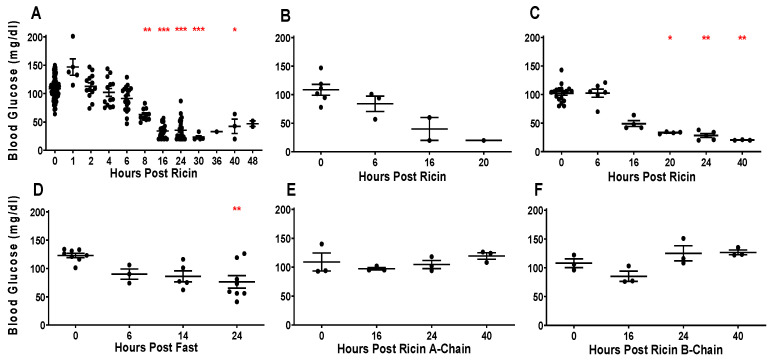
**Hypoglycemic response to parenteral injection of ricin**. Blood glucose of mice at different time points, each dot represents an individual mouse. Blood glucose is shown on the vertical axis and time post injection is on the horizontal axis. Swiss mice were used, except where noted. The mean and standard error for each time point are shown. The Kruskal–Wallis test was performed and the red asterisks above the groups show the results of Dunn’s multiple comparison test vs. the control group: 0.05 (*), 0.002 (**), and <0.001 (***). (**A**) Ip injection of 30 µg/kg of ricin into mice. The mice shown here were used in the experiments described in the remaining figures. (**B**) Iv injection of the same dose of ricin. (**C**) Ip injection of ricin into NSG mice. (**D**) No ricin injection or food given. (**E**) Ip injection of 60 µg/kg ricin A-chain. (**F**) Ip injection of 60 µg/kg ricin B-chain.

**Figure 2 toxins-14-00820-f002:**
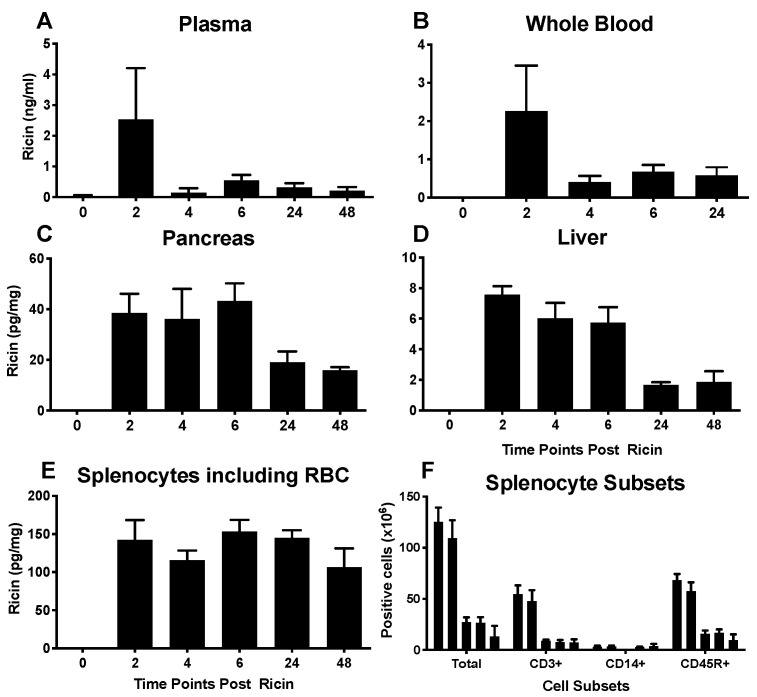
**Localization of ricin post injection**. Mice were injected ip with 30 µg/kg of ricin. Animals were sacrificed at 2, 4, 6, 24, 30 and 48 h post ricin. Each time point represents 3–10 mice. Mean and SEM are shown. (**A**–**E**). Antigen capture ELISA was performed to measure ricin levels in different tissues. Note that the vertical scales are different among the graphs. (**F**). Enumeration of splenocyte subsets at different time points following ricin administration. Shown are the total number of cells per spleen as assayed by flow cytometry. Time points shown by columns from left to right for each cell subset: no ricin, 6, 24, 30 and 48 h post ricin.

**Figure 3 toxins-14-00820-f003:**
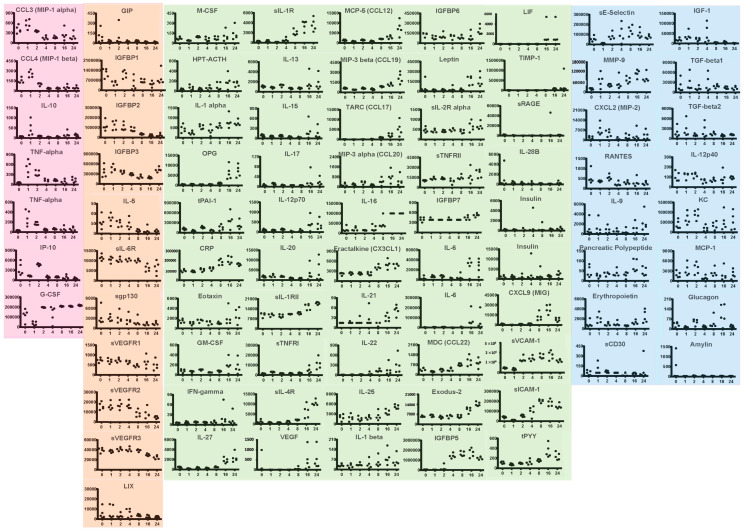
**Multiplex analysis of plasma response to ricin injection**. Groups of five mice were injected ip with 30 µg/kg of ricin. Each dot represents an individual mouse. Samples were obtained prior to and at 1, 2, 4, 8, 16 and 24 h post ricin. Results of multiplex analyses, testing for the cytokines and hormones, are shown above. Time course profiles are grouped based on kinetics of changes. Pink indicates changes within 1–2 h. Orange indicates decrease over time. Green indicates increases over time. Blue has no discernable pattern.

**Figure 4 toxins-14-00820-f004:**
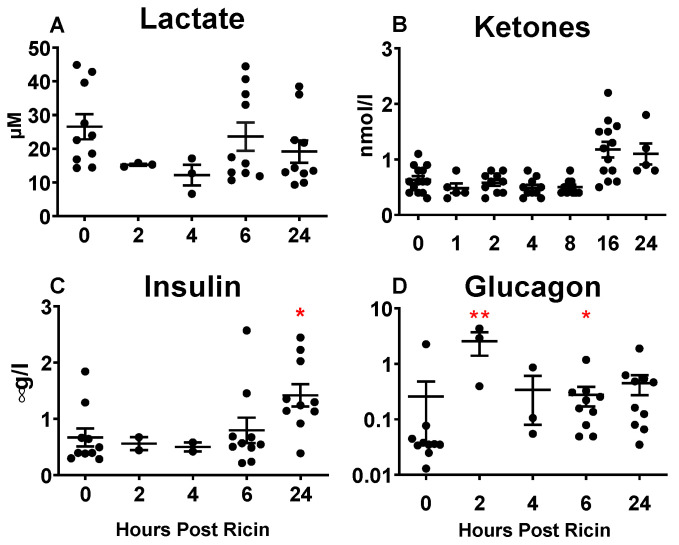
**Systemic metabolic and hormonal response in plasma to ricin injection**. Metabolite or hormone levels were measured in plasma at different time points. Each dot represents an individual mouse. The Kruskal–Wallis test was performed and the red asterisks above the groups show the results of Dunn’s multiple comparison test vs. the control group: 0.05 (*) and 0.002 (**). (**A**) Lactate. (**B**) Ketones. (**C**) Insulin. (**D**) Glucagon.

**Figure 5 toxins-14-00820-f005:**
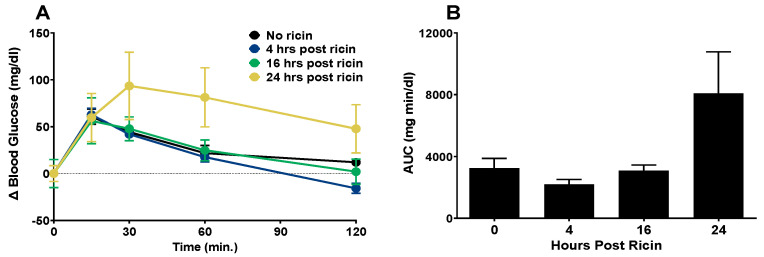
**Glucose tolerance in response to ricin injection**. The mice were injected ip with 30 µg/kg of ricin. A glucose tolerance test was administered at 4, 16 and 24 h post ricin administration. The mice were fasted for 6 h prior to administration of 2 mg glucose/g body weight administered ip. Blood was sampled at 0, 15, 30, 60, and 120 m post glucose injection. (**A**) Change in blood glucose from a normalized baseline blood glucose level. The mean blood glucose levels prior to glucose injection were 103.0 mg/dL, 95.3 mg/dL, 70.3 mg/dL, and 45.3 mg/dL, respectively, for the groups of mice that received no ricin or were injected 4, 16 and 24 h post ricin administration. Each experimental group contained 4–5 mice, and the data represent the mean and SEM for each group. (**B**) Area under the curve calculated from data in panel A. Mean and SEM are shown. One-way ANOVA was performed. The Kruskal–Wallis test with Dunn’s correction for multiple comparison vs. the control group was performed and no statistical significance was observed.

**Figure 6 toxins-14-00820-f006:**
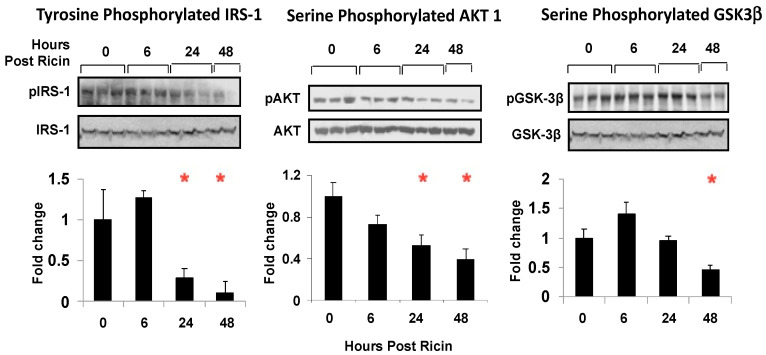
**Insulin signaling post ricin administration.** The mice were injected with ricin 30 µg/kg and sacrificed at the indicated timepoints. The insulin signaling activity was examined in the liver tissue with immunoblotting, shown above the graphs. A “p” prior to the protein name indicates the phosphorylated form. The phosphorylated signal calculated over the non-phosphorylated signal was obtained using Image J. This ratio was then normalized to 1 for the signals of the control group. The signals of mice with ricin exposure were expressed as fold change over the normalized expression of mice with no ricin exposure. One-way ANOVA was performed. Red asterisks above the groups show the results of nested *t* tests with Dunnett’s correction for multiple comparison vs. the control group: 0.05 (*).

**Figure 7 toxins-14-00820-f007:**
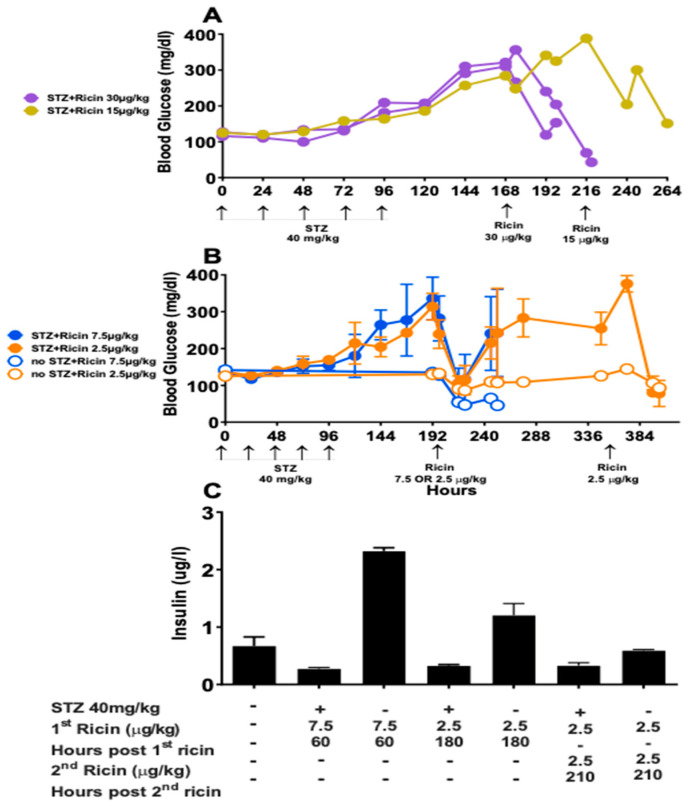
**Blood glucose and insulin following STZ-induced diabetes and ricin injection. Mice received 5 daily doses of STZ at 40 mg/kg to chemically induce diabetes**. Blood glucose was measured daily prior to ricin injection. Post ricin, measurements were taken at 10 a.m. and 4 p.m. daily. (**A**). Each curve represents the blood glucose vs. time of an individual mouse. Two mice were injected with ricin at 30 µg/kg at hour 168, 72 h post last STZ injection. The third mouse was injected with 15 µg/kg of ricin 120 h post STZ. (**B**). The mean and SEM of blood glucose vs. time of groups of three mice are shown. Two treatment groups received STZ and two did not. At hour 192, 96 h post STZ, ricin was injected at 7.5 µg/kg or 2.5 µg/kg. At 276 h post STZ, the groups that received 2.5 µg/kg of ricin received another equivalent dose. (**C**). Plasma insulin was measured in the same mice as in (**B**), mean and SEM are shown.

**Figure 8 toxins-14-00820-f008:**
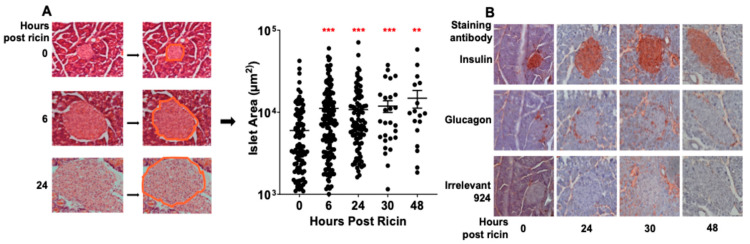
**Islets of Langerhans’ increase in size following ricin injection**. The mice were injected ip with 30 µg/kg of ricin, and they were sacrificed at the indicated time points. The tissue was fixed and histological examination was performed. (**A**) Individual islets were morphologically identified, a region of interest was drawn around the circumference, and the areas were calculated. Only a single section was analyzed per mouse. Each dot represents the area of an identified islet. Mean and SEM are indicated. The Kruskal–Wallis test was performed and the red asterisks above the groups show the results of Dunn’s multiple comparison test vs. the control group: 0.002 (**) and <0.001 (***). (**B**) Immunohistological staining of pancreatic islets. Tissue sections were incubated with Ab to insulin, glucagon, and an irrelevant isotype control followed by alkaline phosphatase conjugated secondary Ab and the substrate aminoethyl carbazole.

**Figure 9 toxins-14-00820-f009:**
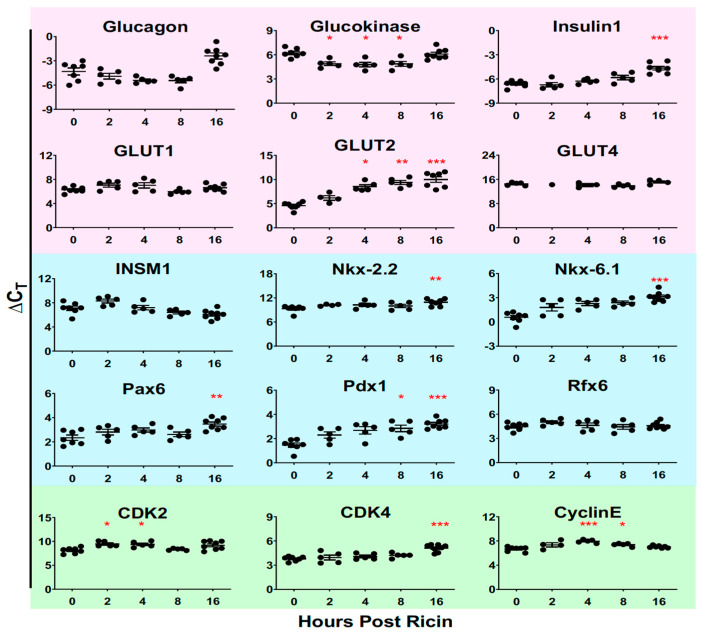
**Real-time PCR quantification of gene expression in isolated islet cells**. The mice were injected with ricin 30 µg/kg and 5 mice were sacrificed at each of the indicated timepoints. Each dot represents an individual mouse. Pancreatic islets were isolated and RNA was extracted. qRT PCR was performed, two replicates were run for each mouse and the mean cycle threshold was determined. ΔCt was calculated by subtracting the cycle threshold of the baseline gene, GADPH, from the cycle threshold of the specified gene. Pink indicates genes involved in glucose regulation, blue endocrine differentiation and green cell cycle regulation. CDK: cyclin-dependent kinase, INSM1: insulinoma-associated antigen-1, GLUT: glucose transporter. The Kruskal–Wallis test was performed and the red asterisks above the groups show the results of Dunn’s multiple comparison test vs. the control group: 0.05 (*), 0.002 (**), and <0.001 (***).

**Figure 10 toxins-14-00820-f010:**
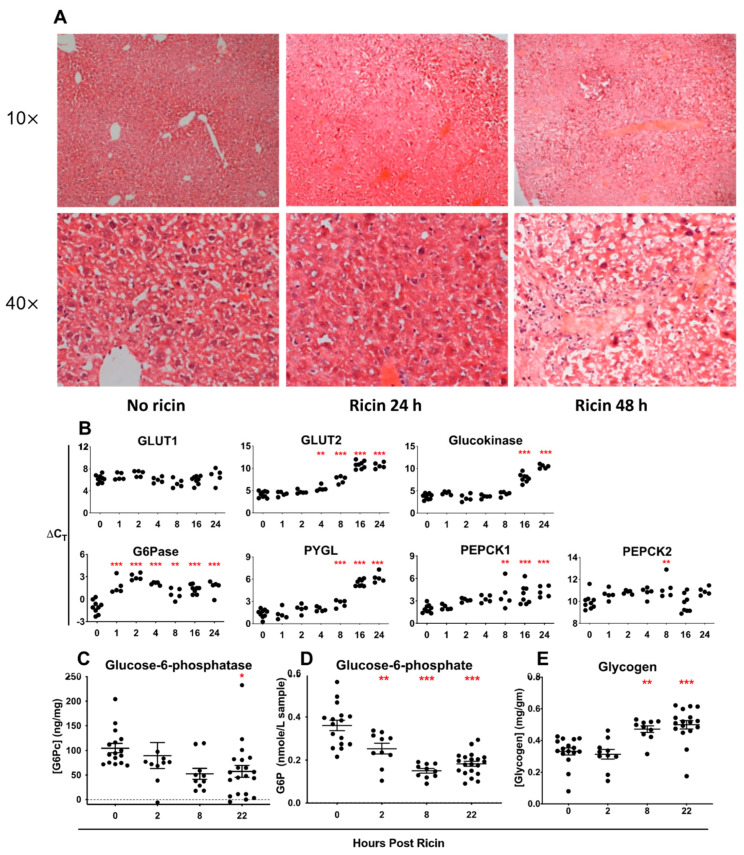
**Histochemical, gene expression, and biochemical profiles in the liver post ricin injection**. The mice were injected with 30 µg/kg of ricin and sacrificed at indicated timepoints. The livers were removed and either immediately frozen or fixed in formalin. (**A**) Fixed tissues were stained with H&E and are shown at 10 and 40 magnification. (**B**) QRT PCR was performed on RNA extracted from the liver. ΔCt was calculated as described previously. GLUT: glucose transporter, G6Pase: glucose-6-phosphatase, PYGL: glycogen phosphorylase for liver, PEPCK: phosphoenolpyruvate carboxykinase. (**C**) ELISA was used to quantify the amount of the catalytic component of G6Pase present in BALB/c mice. (**D**) Glucose-6-phosphate was quantified in BALB/c mice using an enzyme assay. (**E**) Coupled en zyme assay was used to quantify glycogen in BALB/c mice. In panels (**B**–**E**), the Kruskal–Wallis test was performed and the red asterisks above the groups show the results of Dunn’s multiple comparison test vs. the control group: 0.05 (*), 0.002 (**), and <0.001 (***).

**Table 1 toxins-14-00820-t001:** PCR primer pairs used for gene expression studies.

Oligo Name	Sequence	Amplicon
GAPDH	5′-AATGGTGAAGGTCGGTGTGAAC-3′	100
	5′-GTCGTTGATGGCAACAATCTCC-3′	
CDK2	5′-CAGAAATGATTCCCTCCAGTGC-3′	100
	5′-GAACCACGATGAACAGACCAGAG-3′	
CDK4	5′-AAGCGAATCTCTGCCTTCCG-3′	109
	5′-AGGGTTTCTCCACCAAGACTGG-3′	
CyclinE	5′-TTGACCCACTGGACTCTTCACAC-3′	100
	5′-ACAGCAACCTACAACACCCGAG-3′	
Glucagon	5′-GAAGACAAACGCCACTCACAGG-3′	100
	5′-TGGTGTTCATCAACCACTGCAC-3′	
Glucokinase	5′-AAATCCAGGCAAGGACAGGG-3′	100
	5′-AGGGGTAGCAGCAGAATAGGTCTC-3′	
GLUT1	5′-ATCCCAGCAGCAAGAAGGTGAC-3′	100
	5′-GCGTTGATGACACCAGTGTTATAGC-3′	
GLUT2	5′-AGCAATGTTGGCTGCAAACAG-3′	100
	5′-ACTTCGTCCAGCAATGATGAGG-3′	
GLUT4	5′-TGAGAATGACTGAGGGGCAAAAC-3′	100
	5′-GGTAACAGGGAAGAGAGGGCTAAAG-3′	
INSM1	5′-CAGGTGATCCTCCTTCAGGT-3′	102
	5′-CTCTTTGTGGGTCTCCGAGT-3′	
Insulin1	5′-CAGCAAGCAGGTCATTGTTTCAAC-3′	100
	5′-CAAAAGCCTGGGTGGGTTTG-3′	
Nkx-2.2	5′-CCCCATTCCTTTCCTTAAACCC-3′	100
	5′-CCCGCAATTTATGCCACAAAG-3′	
Nkx-6.1	5′-GACTTCGGAGAATGAGGAGGATG-3′	100
	5′-CGATTTGTGCTTTTTCAGCAGC-3′	
Pax4	5′-GCACTGGAGAAAGAGTTTCAGCG-3′	100
	5′-AAACCCTCACCGTGTCTTCAGG-3′	
Pax6	5′-GAAGCGGAAGCTGCAAAGAAATAG-3′	100
	5′-GGCAAACACATCTGGATAATGGG-3′	
Pdx1	5′-GCCCTGAGCTTCTGAAAACTTTG-3′	100
	5′-CCCAGGTTGTCTAAATTGGTCCC-3′	
Rfx6	5′-CACCCTGCATCAAGCCTCTATG-3′	100
	5′-CACAACTGCCACCAAAGAAGTCTC-3′	
Glucose 6-phosphatase	5′-CTGTGGGCATCAATCTCCTCTG-3′	121
	5′-TTGCTGTAGTAGTCGGTGTCCAGG-3′	
Liver Glycogen Phosphorylase (PYGL)	5′-CGACAATGGCTTCTTTTCTCCC-3′ 5′-ACTTGACATAGGCTTCGTAGTCTGC-3′	113

## Data Availability

All data are contained within the manuscript.

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
