# Peer review of "Parenteral Exposure of Mice to Ricin Toxin Induces Fatal Hypoglycemia by Cytokine-Mediated Suppression of Hepatic Glucose-6-Phosphatase Expression"

_toxins, 2022, doi:10.3390/toxins14120820_

Round 1

Reviewer 1 Report

The manuscript entitled “Parenteral Exposure of Mice to Ricin Toxin Induces Fatal Hypoglycemia by Cytokine-Mediated Suppression of Hepatic Glucose-6-Phosphatase Expression” seeks to identify the mechanisms underlying the hypoglycemia caused by ricin intoxication in mice.

The topic is original and of sure interest for “Toxins” readers. The manuscript is well written, the experimental plan designed appropriately and the discussion focused adequately. However, there are some concerns.

Minor points

The following phrase and the related self-citations appear superfluous “Our lab has studied ricin as a pharmacological agent, in anti-HIV immunotoxins [16-18]”

Some interpretations of multiplex analysis are not entirely convincing: GIP e IGFBP1 do not have a clear descending trend; IFN-gamma IL-13, IL-15, IL-17, IL-28B, Insulin and sRAGE do not have a clear ascending trend.

Author Response

See attached file for responses to criticisms.

Reviewer 2 Report

The manuscript entitled ‘Parenteral Exposure of Mice to Ricin Toxin Induces Fatal Hypoglycemia by Cytokine-Mediated Suppression of Hepatic Glucose-6-Phosphatase Expression’ presents interesting insight into toxicity of the ricin, as the vital issue on the way to understand the biological mechanisms of ricin effect in vivo. The studies were focused on the mechanisms associated with the development of hypoglycemia, the one of the observed effects upon parenteral exposure of mice to ricin. Authors considered two aspects of the response to parenteral administration of ricin, namely inflammation and glucose homeostasis via the pancreatic/hepatic axis; they suggest that the observed hypoglycemia caused by ricin injection induces an inflammatory response, with prominent induction of TNF-α production and they linked this phenomenon to downregulation of the G6Pase expressions, which prevents hepatic gluconeogenesis, and development of systemic hypoglycemia.

The manuscript is well written, with emphasis to metabolic outcomes associated with ricin action on the mouse organism, providing semi-metabolic analysis of mouse responses to ricin administration, being highly focused on hepatic and pancreatic regulators of glucose metabolism. However, despite the overall benefit which is provided by the presented manuscript, several issues should be refined to provide audience with more clear data. The publication is centered on ricin metabolism and Authors should provide the audience with a short mechanistic view of the ricin mode of action, explaining the primary target of the ricin which is the GTPase Associated Center with the Sarcin-Ricin Loop - SRL as a main object; thus some important data should be provided such as docking site for the ricin (PMID: 28606931, PMID: 28717148); importantly, Authors provided wrong data about the ricin action, stating that: ‘… the A chain enzymatically cleaves the large ribosomal unit, halting protein synthesis …’ - ricin does not cleave the large ribosomal subunit, but catalyzes depurination of a single adenine base on the SRL; also, the inhibition of protein synthesis by the ricin was shown only in vitro, and recent data indicate that the ricin impact on the translational machinery is more complex in vivo, and does not involve only inhibition of protein synthesis per se (PMID: 31518597). Authors should also pay more attention to apoptosis issue, that is the main cause of cell death, induced by the ricin and they should provide more information about physiological outcomes of the ricin intoxication (PMID: 35024573); especially in the discussion section which is primarily dominated by glucose metabolism, which is probably secondary effect of ricin action. Next, Authors should give profound explanation for the used ricin dose, especially the LD-LO, LD50 and LD99 should be provided. Next, Authors used the commercially available ricin, the quality of the protein fraction should be shown, for example by SDS-PAGE. Next, the data in Fig. 4 should be integrated, for better data presentation; at the moment it is difficult to follow the ‘richness’ of the data. Next, the pancreatic and liver histology and function should be improved, thus data about apoptosis should be shown, especially after 24hr of ricin treatment, apoptotic cells may dominate; also, Authors are showing the functional data in relation to the expression level of several mRNA for protein involved in glucose metabolism using RT-PCR; however, without analysis toward definition of apoptosis, the RT-PCR approach provide little data, because we do not know about metabolic state of the cells, and whether mRNA fluctuation are simply apoptotic related or there are another metabolic background. Additionally, mRNA level and level for particular protein are not frequently corelated, thus changes in mRNA level may not corelate with actual protein level in the cells, thus RT-PCR might be misleading; therefore the western blotting should be the method of chois. Next, I have reservation to the ELISA analysis of G6Pase level, because ‘specific’ antibodies very frequently may give cross-reaction, thus the western-blotting would give much better reliable analysis, as it is very important analysis for the final conclusion of the Authors.

Author Response

(The authors gave the same response as above.)

Reviewer 3 Report

The ms ”Exposure of Mice to Ricin Toxin Induces Fatal Hypo glycemia by Cytokine-Mediated Suppression of Hepatic Glucose-6-Phosphatase Expression” exposes the metabolic and systemic response ricin toxin-induced hypoglycemia in mice after parenteral administration of the toxin. In relationship to biological warfare and bio-terrorism authors were interested in an early diagnostic of ricin exposure and propose the toxin-induced hypoglycemia as a possible candidate to be a ricin-specific biomarker.

In general, authors describe the cellular response on ricin exposure. The  experimental approach is very straightforward, and the obtained results are descriptive. Interpretation of the data is based on general immunology and a plausible mechanism and explanation of the results is exposed by authors, but at the same time authors remain cautious and discuss that the obtained results are questionable. It is not clear if there is a direct link between cause and effect and how this is related to the presence of inflammatory cytokines. Moreover, it is discussed whether these observations are valid when referred to humans as mice do not necessarily respond in the same way. Though it is ethically complicated experiments should be performed in humans, or at least check the cellular response of human cells as e.g. hepatocytes. It would be interesting to know (in the murine system) if the if the clearance of the toxin (as far as possible) restores normal body homeostasis or if there is any other possible treatment of patients that have been exposed to ricin.

Though the paper is rather descriptive and does not provide direct answers I have no inconvenience to recommend its publication.

Author Response

(The authors gave the same response as above.)

Reviewer 4 Report

I found this paper very interesting and well constructed and quite complex. The interaction between ricin toxin, inflammation and hypoglycemia was comprehensively defined and studied. A lot of data presented in the paper and the use of a lot of abbreviations makes paper difficult to follow. Thus, I have a feeling that list of abbreviations would be beneficial to the paper. Also in conclusions some simplified scheme summarizing the presented studies woulkd be acknowledged. This could be in part similar to graphical abstract of the paper presented by Zhou et. al in Mol. Cell Endocrinol

Author Response

(The authors gave the same response as above.)

Round 2

Reviewer 2 Report

suitable